# Examining the effectiveness of the Gateway conditional caution on health and well-being of young adults committing low-level offences: a randomised controlled trial

Alex Mitchell ,[1] Alison Booth ,[1] Sara Morgan,[2] Inna Walker,[2] Caroline Chapman,[3] Megan Barlow-Pay,[2] Ann Cochrane,[1] Emma Filby,[1] Jenny Fleming,[4] Catherine Hewitt,[1] James Raftery ,[5] David Torgerson,[6] Lana Weir,[2] Julie Parkes[7]

For numbered affiliations see end of article.

**Correspondence to**
Alex Mitchell;
alex.mitchell@york.ac.uk

## ABSTRACT

**Background** Young adults who commit low-level offences commonly have a range of health and social needs and are significantly over-represented in the criminal justice system. These young adults may need to attend court and potentially receive penalties including imprisonment. Alternative routes exist, which can help address the underlying causes of offending. Some feel more should be done to help young adults entering the criminal justice system. The Gateway programme was a type of out-of-court disposal developed by Hampshire Constabulary, which aimed to address the complex needs of young adults who commit low-level crimes. This study aimed to evaluate the effectiveness and cost-effectiveness of the Gateway programme, issued as a conditional caution, compared with usual process.

**Methods** The Gateway study was a pragmatic, parallel-group, superiority randomised controlled trial that recruited young adults who had committed a low-level offence from four sites covering Hampshire and Isle of Wight. The primary outcome was mental health and well-being measured using the Warwick-Edinburgh Mental Well-being Scale. Secondary outcomes were quality of life, alcohol and drug use, and recidivism. Outcomes were measured at 4, 16 and 52 weeks postrandomisation.

**Results** Due to issues with retention of participants and low data collection rates, recruitment ended early, with 191 eligible participants randomised (Gateway 109; usual process 82). The primary outcome was obtained for 93 (48.7%) participants at 4 weeks, 93 (48.7%) at 16 weeks and 43 (22.5%) at 1 year. The high attrition rates meant that effectiveness could not be assessed as planned.

**Conclusions** Gateway is the first trial in a UK police setting to have a health-related primary outcome requiring individual data collection, rather than focusing solely on recidivism. We demonstrated that it is possible to recruit and randomise from the study population, however follow-up rates were low. Further work is needed to identify ways to facilitate engagement between researchers and vulnerable populations to collect data.

**Trial registration number** ISRCTN11888938.

## STRENGTHS AND LIMITATIONS OF THIS STUDY

⇒ The planned pragmatic trial was robustly and transparently planned and involved close collaboration between a wide range of stakeholders.
⇒ We were not able to assess effectiveness of the Gateway intervention due to low data collection rates.
⇒ Our work on this trial has provided a robust benchmark for attrition which will help guide future health related trials in the police setting and with 18–24 years old committing low level crimes.

## BACKGROUND

Young adults who commit low-level offences commonly have a range of health and social needs, making them vulnerable to mental health problems.[1–3] These young offenders are more likely to come into contact with the police both as suspects and victims of crime and are significantly over-represented in the criminal justice system, accounting for approximately one-third of police, probation and prison caseloads.[4] According to statistics from Hampshire Constabulary (HC) for 2018/2020, the five main low-level offence categories for adults aged between 18 and 24 where formal action was taken by the police are possession of drugs, violence, shoplifting, criminal damage and public order offences. Young adults who have been investigated for a suspected low-level offence, may need to attend court and, if convicted, face penalties such as prison.

More could be done to help young adults entering the criminal justice system, for example, via court diversion programmes. Diversion is a process whereby an accused

BMJ

person is formally moved into a programme in the community, such as an out-of-court community-based intervention, instead of a court summons.[5] In the UK, a number of police forces are exploring the use of out-of-court disposals (OOCD, an alternative to a court summons) among 18–24 years old involved in less serious offending.[6–9] The aim is to divert the young adult away from their offending behaviour through a rehabilitative path.[10]

The Gateway programme was issued as a novel form of conditional caution, where release from custody comes with mutually agreed conditions. Gateway was conceived by HC as a culture-changing initiative that sought to address the complex needs of adults aged 18–24 years who commit low-level crimes. However, HC recognised the need for evidence on the effectiveness of Gateway and were keen on an evaluation of its effectiveness in relation to a wider set of outcomes beyond recidivism, with a particular focus on health and well-being of young people.

The aim of this study was therefore to evaluate the effectiveness and cost-effectiveness of the Gateway programme issued as a conditional caution, compared with usual process (a court appearance or a different conditional caution), in relation to health and well-being of its clients.

## METHODS

A summary of the study methods is given here; full details are available in the published protocol paper,[11] and the protocol available online (https://www.fundingawards.nihr.ac.uk/award/16/122/20).

### Study design

The Gateway study was a pragmatic, multicentre, superiority randomised controlled trial (RCT) that compared two groups of young adults who had committed a low-level offence. Participants were randomised to either the Gateway conditional caution (intervention) or disposal as usual to a court summons or a different conditional caution (usual process). An economic evaluation was planned. A qualitative evaluation of the impact of the intervention on participants and other stakeholders will be reported elsewhere.

Participants were recruited from four sites (Southampton, Portsmouth, Isle of Wight and Basingstoke Police Stations), covering the whole of Hampshire and Isle of Wight. Follow-up was carried out at 4 weeks, 16 weeks and 1 year postrandomisation.

### Participants

Participants were eligible if they were aged 18–24 years, resided in the Hampshire and Isle of Wight area, were anticipated to give a guilty plea and there was sufficient evidence to provide a realistic prospect of conviction, and it was in the public interest to prosecute or offer a conditional caution to the suspect. Exclusion criteria included serious and indictable only offences, and those involving domestic or sexual violence, knives, hate, serious injury, drink-driving, breach of offence orders and any serious previous conviction. Those needing an interpreter or having a previous Gateway caution were excluded.

### Recruitment

By law the police must know the destination for an offender at the time of disposal, that is, when the outcome of the investigation is administered. As the intervention was one of the disposal options, randomisation had to take place at the time of disposal. HC investigators were trained to identify, recruit and randomise participants, an approach that had previously been used.[12]

It was not felt appropriate for police investigators to obtain full consent because of the potential risk of coercion, nor was it practical, given the timelines. We therefore developed a two-stage consent procedure. During processing in custody, investigators identified potentially eligible participants and discussed with them the Gateway caution. For legal reasons, the Gateway caution was initially offered as a disposal option independently of the study. If interest was shown, the young person was then informed about the study. A Gateway Caution information leaflet (produced by HC independently of the study) and a study leaflet with a link to an explanatory video were shared. Potential participants were made aware that further details about the study would be provided by a researcher and that they could withdraw from the study at any time without giving a reason. If the young person was interested in the opportunity to receive Gateway and take part in the study, the investigator obtained stage 1 consent. This allowed HC to share their contact details with the University of Southampton researchers and gave York Trials Unit (YTU) researchers access to their police record for demographics such as age, gender and ethnicity and offending history, trigger offence and any subsequent reoffending. This process precluded the collection of baseline outcome data.

Some participants were out of custody when it was decided the arrest criteria had been met and/or Gateway was suitable. For these participants, verbal consent was obtained over the telephone and randomisation undertaken at that time. It was therefore possible that the subsequent in person disposal for some of these participants could occur several weeks after randomisation depending on when the in-person disposal could be arranged. Study procedures continued as per protocol.

Ahead of the week 4 data collection time point, the researchers attempted to contact participants by telephone, text, email and/or post to arrange an interview. Once arranged, the stage 2 participant information sheet was emailed or posted to the participant. At the interview the researcher went through the information sheet providing explanations as required. If the patient consented, data collection could occur at the same interview or on a subsequent day. To maximise data collection, if a participant took part in the week 16 interview having

not taken part at week 4, verbal consent was obtained at that point.

## Randomisation and blinding

Police officers and investigators (hereafter referred to as investigators) coming into contact with potential participants were offered opportunities to undergo related training prior to the start of the study, as well as once the study was live, which was aimed mainly at new staff and as refresher training. Potential participants were screened using an online eligibility tool hosted by Alchemer and developed by HC in discussion with YTU. Eligible young people were consented by investigators using a guidance script developed jointly by HC and the research team. Consenting participants were randomised using a 1:1 allocation ratio with simple randomisation. Researchers involved in consenting and collecting data from participants were blind to allocation. It was not possible to blind participants due to the nature of the intervention.

## Intervention and usual care

The Gateway conditional caution was a police-led intervention delivered using a multi-agency approach.

The Gateway intervention consisted of three compulsory parts.
1. Within 3–5 working days of their disposal, the participant met with a Gateway navigator for a needs assessment. The navigator then assisted the young adult into the appropriate services, including Gateway partner agencies (eg, housing, alcohol, drug and mental health services). The navigators also undertook midway and final assessments and provided mentoring throughout the programme. The Gateway navigators were trained support workers, provided by a third sector organisation, No Limits, and by Southampton City Council.
2. Attendance at two LINX workshops run by The Hampton Trust (HT) aimed to assist young adults in the development of cognitive and affective empathy and prevent reoffending. These were delivered between weeks 2–3 and 5–6 postrandomisation.
3. Undertaking not to reoffend during the 16 weeks of the conditional caution.

Additional conditions could also be added at the discretion of the supervising officer approving the disposal destination. If a participant reoffended during the period of their caution, the HC Gateway Team could use their discretion when deciding whether a breach had occurred. If a participant was considered to have breached the terms of the caution, they were withdrawn from the Gateway intervention, and the original investigator considered whether to prosecute the participant for the original offence. Participants who breached their Gateway Conditional Caution continued to be approached for data collection.

Participation in Restorative Justice could be requested by the victim, but this was not part of the standard Gateway caution.

Usual process consisted of either a different conditional caution or the participant being charged to appear in court. Examples of conditions attached to the usual process caution include apology letters, victim awareness courses, drug or alcohol diversion courses, fines and compensation.

## Changes to the intervention and usual process as a result of the COVID-19 pandemic

In response to government restrictions, on 22 March 2020 HC halted all conditional caution activities that involved face-to-face interaction. The in-person nature of the Gateway intervention meant delivery modes had to change. The navigators modified their practice to undertake needs assessments and meetings with clients by telephone as standard. The content and purpose of the initial needs assessment and subsequent contact remained the same. The HT modified the workshops to be delivered one-to-one over the telephone. The principles and key elements of the workshops were maintained but reduced in length from 10 hours to 2 hours. Face-to-face working returned in May 2021, where appropriate and risk assessed.

In terms of usual care, simple cautions and conditional cautions with conditions relating to fines, compensation and apology letters continued to be issued; court proceedings were halted. However, as the intervention was unavailable, recruitment was halted on 23 March 2020. In August 2020, HC restarted all conditional cautions, including Gateway.

## Outcomes

The primary outcome was the Warwick-Edinburgh Mental Well-being Scale (WEMWBS), which measures mental health and well-being. The WEMWBS consists of 14 items, each with a 5-point scale. The total score ranges from 14 to 70, with a higher score indicating a higher level of health and well-being.

The patient-reported secondary outcomes were the Short Form-12 (SF-12) mental and physical components, Alcohol Use Disorders Identification Test (AUDIT) and Adolescent Drug Involvement Scale (ADIS) scores. The ADIS also has an additional section on the use of different types of drugs that enables a score titled the Index of Multiple Drug Use to be scored. This was not a study outcome but is reported in the results. Secondary outcomes measuring recidivism 1 year postrandomisation were the total number of police records management system (RMS) incidents, the total number of RMS incidents resulting in being charged or cautioned, the total number of police national computer (PNC) convictions, whether the participant was charged with a summary or either-way offence and whether the participant was charged with an indictable only offence. In the statistical analysis plan it was originally stated the first two recidivism outcomes would be the total number of RMS incidents plus the total number of PNC convictions up to 1 year postrandomisation and the total number of

RMS incidents resulting in being charged or cautioned plus the total number of PNC convictions. However, on receipt of the RMS and PNC data we found that a single offence could be classed as both an incident in the RMS data and a conviction in the PNC data, and hence would lead to double counting when deriving these two recidivism outcomes. It was therefore decided to separate out the number of PNC convictions and report it as its own outcome.

## Patient and public involvement

Patient and public involvement (PPI) was embedded early on with the help of partners The HT. Meetings with young adults on an HT programme explored various aspects of the study, including importance, acceptability and feasibility. The groups fed back in detail around the logistics of the study: the process around consent and randomisation; ways to manage challenges following up the control arm; and opinion on assessment forms.

Once the study was underway, the PPI lead worked with partners to involve young adult representatives who had been through the Gateway programme and those who had been through the 'usual process'. Consultation and input from these service users provided a clear understanding of the challenges and benefits that participants with and without prior experience of the criminal justice system might face. These PPI representatives worked closely with the PPI lead to develop consent forms, PISs, and initial information leaflets, plan recruitment strategies and consider the most effective ways of arranging interviews and qualitative work.

There were two public representatives on the Study Steering Committee/Data Monitoring and Ethics Committee (SSC/DMEC). An ex-offender, working for Hampshire Youth Offenders Team as a peer mentor and support worker; and a victim advocate, working for a charity for victims of crime. They represented the voice of the service users and victims at Steering Group meetings, helping the group reflect on the realities of delivering the programme from the user perspective, reminding the group of some of the vulnerabilities and needs of this population, and ensuring the views of victims were considered.

These two representatives also worked closely with the study PPI lead, providing strategic input, advice and guidance throughout, with a particular focus on the logistics of getting the project underway, reviewing and adapting the protocol. The idea of a recruitment video was conceived by the ex-offender public representative, and the content was cocreated with them.

Using links established through a local outreach programme, community leaders and members of the public were consulted. We worked closely with these individuals to ensure we understood the concerns and attitudes of the wider community. Additionally, they were able to provide input to public facing documentation and materials.

## Statistical analysis

It has been suggested that a change of three or more points on the WEMWBS is likely to be important to individuals, although different statistical approaches provide different estimates ranging from three to eight points (WEMWBS user guide[13]). Estimates of the SD also vary between 6 and 10.8,[14] with a pooled estimate of 10 across all studies. Assuming 90% power, 5% statistical significance, a minimal clinically important difference of 5 points on the WEMWBS and an SD of 10, 266 participants were required. Preliminary figures from The Hampton Trust's Raising Awareness of Domestic Abuse in Relationships intervention suggested a drop-out rate of approximately 15%. Assuming a conservative 20% attrition rate, we aimed to recruit and randomise 334 participants.

Analyses were conducted in Stata V.17 (StataCorp; College Station, TX, USA) and followed a prespecified statistical analysis plan (SAP) approved by the SSC and DMEC prior to the completion of data collection.

Version 1.0 of the SAP outlined the planned analyses to assess the effectiveness of the Gateway intervention, however, poor retention and data collection rates made this unfeasible. Version 1.1 of the SAP removed all reference to formal hypothesis testing and outlined purely descriptive analyses.

Continuous measures were summarised using counts, mean, SD, median, IQR, minimum and maximum. Categorical measures were summarised using counts and percentages. All participants were analysed according to their randomised group, unless otherwise stated. The flow of participants from eligibility and randomisation to follow-up and analysis of the trial was presented in a Consolidated Standards of Reporting Trials flow diagram.[15] Reasons for ineligibility and non-consent were given. The number of withdrawals and reasons for withdrawal at each time point were summarised descriptively by randomised treatment group. Participant demographics were summarised descriptively by randomised treatment group, both for all participants randomised and participants who provided the primary outcome data for at least one timepoint. No formal statistical comparisons were undertaken between groups.

For those who received Gateway, the number of LINX workshops attended, delivery of LINX workshops, contacts attempted by the navigator, successful contacts made by the navigator and total duration of successful contacts were summarised descriptively. For participants who were cautioned, the conditions attached to each caution were summarised descriptively by whether the participant received the Gateway conditional caution or a different caution.

The primary, secondary and exploratory outcomes were summarised descriptively at each timepoint by randomised group.

Intervention compliance was defined as both minimal compliance and full compliance. Minimal compliance was met when the participants engaged with their navigator at the initial, midway and final assessments, attended

the two LINX workshops and had not been breached for reoffending during the duration of the conditional caution. Full compliance was met when the conditions for minimal compliance were met, and in addition the participant engaged with external agencies organised by the navigator.

The number and proportion of participants informed of their disposal decision after their 4-week follow-up was due, was presented by randomised treatment group. The number of days between randomisation and date of disposal were summarised descriptively, alongside whether the participant attended their 4-week follow-up. The number and proportion of participants in the intervention group who violated the condition to reoffend was presented. For these participants, the number for whom discretion was considered before taking the decision to breach was reported.

## RESULTS

Due to issues with retention of participants and data collection rates, recruitment ended on 13 December 2021, and data was collected for participants due up until 31 March 2022.

Between 1 October 2019 and 13 December 2021 345 potentially eligible young people were screened, of which 298 (86.4%) were eligible. Of the 298 eligible, 106 (35.6%) did not consent to the study. Of these, 77 (72.6%) refused the study but accepted the Gateway caution; 5 (4.7%) refused the Gateway caution; 2 (1.9%) ran out of prosecution time; and 2 (1.9%) were missed by the recruiting investigator (reason unknown). There were 20 (18.9%) for whom the reason for non-consent is unknown. In total, 192 (64.4%) participants were recruited and randomised. One participant was randomised in error, which led the custody sergeant to non-randomly assign the participant. This participant is excluded from all further analyses, meaning 191 participants were randomised and included in the analyses (Gateway 109; usual process 82; figure 1).

The mean age of participants was 20.8 years (range 18.1–24.8) and 144 (78.7%) were male (table 1). The median total number of RMS incidents involved in 1 year prerandomisation was 6 (IQR 3–13), with 57 (31.5%) participants involved in an RMS incident that led to a caution or charge during this period. Baseline characteristics of the randomised participants were generally balanced between groups, except for small imbalances in gender and highest level of education. For participants who provided a valid WEMWBS score, there was an imbalance in the proportion of participants previously convicted that was larger than the imbalance observed in all randomised participants.

Of the 109 participants randomly assigned Gateway, 104 (95.4%) received Gateway with 4 of the remaining 5 receiving a standard caution. Of the 81 (98.8%) participants who were randomly assigned to and received usual process, 76 (93.8%) entered the study via the caution route that is, received a different conditional caution.

There were 18 (17.1%) who received a Gateway caution with the additional condition of providing compensation, while 5 (4.8%) were required to write a letter of apology the victim. Of those who received a simple or conditional caution, the most common conditions attached were compensation (n=20; 25.0%), attending a drug diversion course (n=16; 20.0%) and attending a victim awareness course (n=14; 17.5%).

Of the 105 participants who received Gateway, data on number of LINX sessions attend was received for 101 (96.2%), of which 88 (87.1%) attended both sessions, 1 (1.0%) attended one session, 8 (7.9%) did not attend any sessions, while 4 (4.0%) could not attend due to the COVID-19 pause. Of those who attended at least one workshop, 45 (56.3%) attended a face-to-face workshop while 35 (43.8%) had the workshop delivered via the telephone. The median number of successful contacts made by the navigator to the participant was 19 (IQR 15–31). For each participant the total duration of successful contacts was calculated, the median of which was 626.5 min (IQR 380–978). Further information on the delivery of Gateway and usual process is presented in online supplemental appendix A.

At the primary endpoint of 1 year postrandomisation, 43 (22.5%) case report forms (CRFs) were returned (Gateway 27, 24.8%; usual process 16, 19.5%) (figure 1). At 4 weeks postrandomisation 94 (49.2%) CRFs were returned (Gateway 58, 53.2%; usual process 36, 43.9%) while at 16 weeks postrandomisation 95 (49.7%) (Gateway 56, 51.4%; usual process 39, 47.6%). The WEMWBS, SF-12, AUDIT and ADIS data for one participant in the Gateway group was excluded at week 4 due to the questionnaire being completed too early. At week 16 the data for two participants in the Gateway group were excluded due to the questionnaires being completed too late.

Valid participant-reported outcome data was provided by 96 (50.3%) participants at the 4-week follow-up, 93 (48.7%) participants at the 16-week follow-up and 43 (22.5%) participants at the 1-year follow-up (Gateway 56, 51.4%; usual process 39, 47.6%). Descriptive summaries of the primary and secondary outcomes are provided in tables 2 and 3, respectively.

There were 129 (67.5%) participants who had reached the 1-year follow-up before their RMS data was extracted by HC on 23 June 2022, while 125 (65.4%) reached the 1-year follow-up before their PNC data was extracted. Ten participants who withdrew before or after stage 2 consent, declined stage 2 consent or lost mental capacity did not have their RMS and PNC data reported. Of the 32 participants in the Gateway group who had been in the study less than 1 year, 2 (6.3%) had been charged with a summary or either-way offence, while of the 24 participants in the usual process group, 2 (8.3%) had been charged. For the 56 participants who had been in the study less than 1 year, the mean time between date of randomisation and date of data extraction was 286.9 days (SD 56.7 days). Table 4 gives descriptive summaries of the recidivism outcomes.

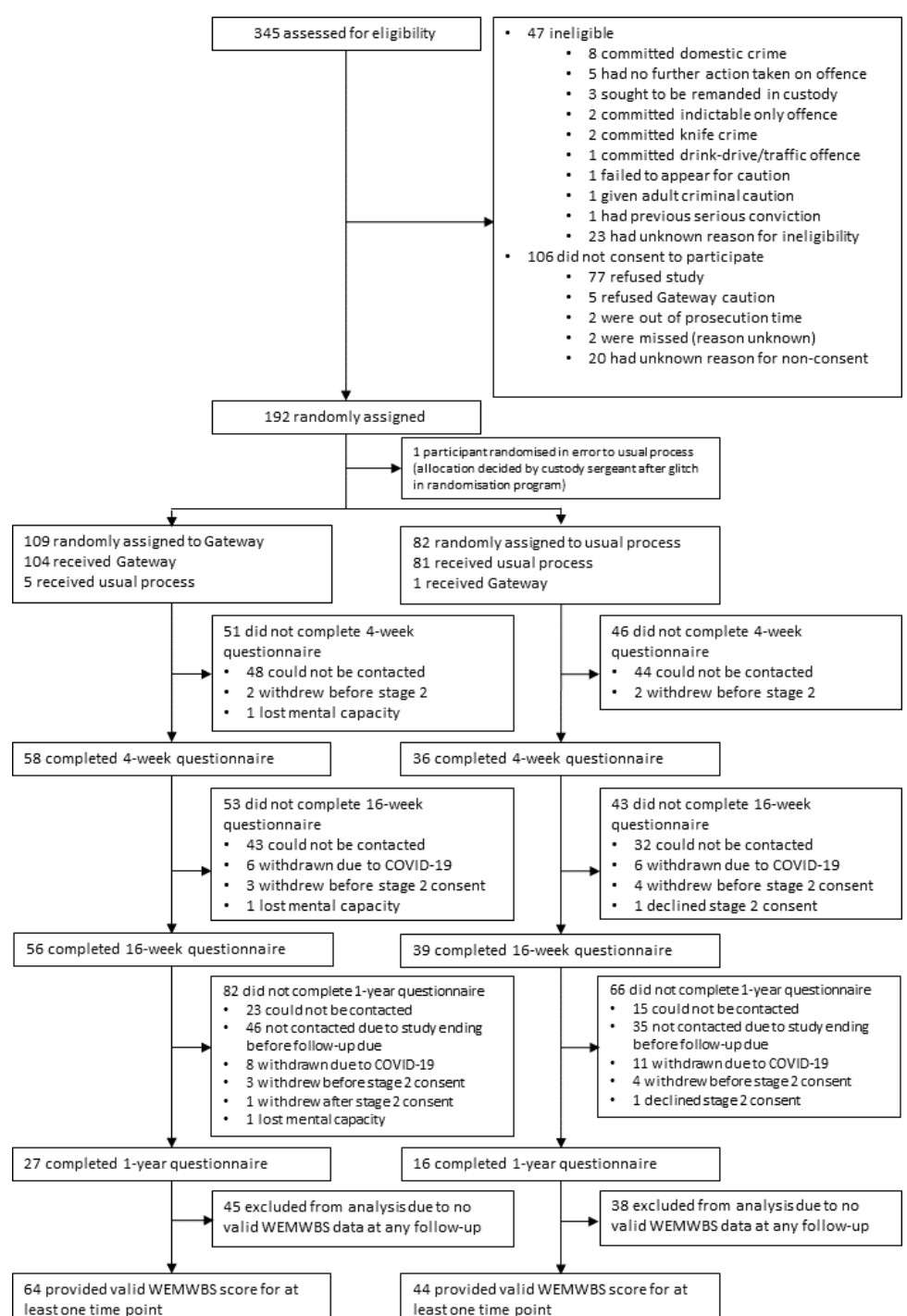

**Figure 1** Consolidated Standards of Reporting Trials diagram demonstrating the progression of participants through the trial. WEMWBS, Warwick-Edinburgh Mental Well-being Scale.

Of the 105 participants randomly allocated to the Gateway conditional caution who did not withdraw before stage 2 or withdraw stage 2 consent, 81 (77.1%) met the definition for minimal compliance. Thirteen participants did not meet minimal compliance due to not attending the two LINX sessions, six did not meet minimal compliance due to breaching the condition to not reoffending during the period of the caution and five were given usual process despite being randomly assigned to the Gateway conditional caution.

No participants were withdrawn from the Gateway conditional caution because they failed to engage with referral agencies identified by the navigator, therefore the number of participants meeting full compliance was 81 (77.1%).

Of the 191 randomised participants, 15 (7.9%) were informed of their disposal decision after their 4-week follow-up was due (Gateway 12, 11.1%; usual process 3, 3.7%; see online supplemental appendix B).

Of the 105 participants who received the Gateway conditional caution who did not withdraw before stage

**Table 1** Participant characteristics presented by allocated group, for all randomised participants and all randomised participants who provided a valid WEMWBS score for at least one timepoint

| | Randomised participants (n=191) | | | Provided valid WEMWBS for at least one timepoint (n=108) | | |
|---|---|---|---|---|---|---|
| | Gateway conditional caution (n=109) | Usual process (n=82) | Total (n=191) | Gateway conditional caution (n=64) | Usual process (n=44) | Total (n=108) |
| **Age at randomisation** | | | | | | |
| Number with data, n (%) | 105 (96.3) | 78 (95.1) | 183 (95.8) | 64 (100) | 44 (100) | 108 (100) |
| Mean (SD) | 20.8 (2.0) | 20.7 (1.9) | 20.8 (1.9) | 20.7 (2.0) | 20.7 (1.7) | 20.7 (1.9) |
| Median (IQR) | 20.3 (19.3, 22.5) | 20.4 (19.3, 21.6) | 20.4 (19.3, 22.0) | 20.2 (19.0, 22.3) | 20.5 (19.4, 21.4) | 20.3 (19.3, 21.6) |
| Min, Max | 18.1, 24.8 | 18.1, 24.8 | 18.1, 24.8 | 18.1, 24.7 | 18.1, 24.7 | 18.1, 24.7 |
| **Gender, n (%)** | | | | | | |
| Number with data, n (%) | 105 (96.3) | 78 (95.1) | 183 (95.8) | 64 (100) | 44 (100) | 108 (100) |
| Male | 87 (82.9) | 57 (73.1) | 144 (78.7) | 51 (79.7) | 32 (72.7) | 83 (76.9) |
| Female | 18 (17.1) | 21 (26.9) | 39 (21.3) | 13 (20.3) | 12 (27.3) | 25 (23.1) |
| **Marital status, n (%)** | | | | | | |
| Number with data, n (%) | 66 (60.6) | 44 (53.7) | 110 (57.6) | 64 (100) | 44 (100) | 108 (100) |
| Single | 62 (93.9) | 38 (86.4) | 100 (90.9) | 60 (93.8) | 38 (86.4) | 98 (90.7) |
| Living with partner | 4 (6.1) | 5 (11.4) | 9 (8.2)) | 4 (6.2) | 5 (11.4) | 9 (8.3) |
| Married | 0 (0) | 1 (2.3) | 1 (0.9) | 0 (0) | 1 (2.3) | 1 (0.9) |
| **Ethnicity, n (%)** | | | | | | |
| Number with data, n (%) | 104 (95.4) | 77 (93.9) | 182 (94.8) | 63 (98.4) | 44 (100) | 108 (100) |
| White North European | 96 (91.4) | 75 (96.2) | 170 (93.4) | 58 (90.6) | 44 (100) | 102 (94.4) |
| Black | 5 (4.8) | 2 (2.6) | 7 (3.8) | 3 (4.7) | 0 (0) | 3 (2.8) |
| Asian | 2 (1.9) | 1 (1.3) | 3 (1.6) | 1 (1.6) | 0 (0) | 1 (0.9) |
| White South European | 1 (1.0) | 0 (0) | 1 (0.5) | 1 (1.6) | 0 (0) | 1 (0.9) |
| **Highest level of education, n (%)** | | | | | | |
| Number with data, n (%) | 66 (60.6) | 44 (53.7) | 110 (57.6) | 64 (100) | 44 (100) | 108 (100) |
| No qualifications | 14 (21.2) | 3 (6.8) | 17 (15.5) | 14 (21.9) | 3 (6.8) | 17 (15.7) |
| 1–4 GCSEs | 20 (30.3) | 8 (18.2) | 28 (25.5) | 20 (31.3) | 8 (18.2) | 28 (25.9) |
| More than 5 GCSEs | 13 (19.7) | 11 (25.0) | 24 (21.8) | 13 (20.3) | 11 (25.0) | 24 (22.2) |
| Apprenticeship | 2 (3.0) | 5 (11.4) | 7 (6.4) | 2 (3.1) | 5 (11.4) | 7 (7.5) |
| 2 or more A-levels | 17 (25.8) | 15 (34.1) | 32 (29.1) | 15 (23.4) | 15 (34.1) | 30 (27.8) |
| Bachelor's degree or higher | 0 (0) | 2 (4.5) | 2 (1.8) | 0 (0) | 2 (4.5) | 2 (1.9) |

Continued

**Table 1** Continued

| | Randomised participants (n=191) | | | Provided valid WEMWBS for at least one timepoint (n=108) | | |
|---|---|---|---|---|---|---|
| | Gateway conditional caution (n=109) | Usual process (n=82) | Total (n=191) | Gateway conditional caution (n=64) | Usual process (n=44) | Total (n=108) |
| **IMD quintile (1=most deprived, 5=least deprived), n (%)** | | | | | | |
| Number with data, n (%) | 94 (86.2) | 72 (87.8) | 166 (86.9) | 58 (90.6) | 42 (95.5) | 100 (92.6) |
| 1 | 21 (22.3) | 20 (27.8) | 41 (24.7) | 14 (24.1) | 14 (33.3) | 28 (28.0) |
| 2 | 25 (26.6) | 17 (23.6) | 42 (25.3) | 14 (24.1) | 9 (21.4) | 23 (23.0) |
| 3 | 15 (16.0) | 14 (19.4) | 29 (17.5) | 9 (15.5) | 8 (19.0) | 17 (17.0) |
| 4 | 16 (17.0) | 7 (9.7) | 23 (13.9) | 9 (15.5) | 4 (9.5) | 13 (13.0) |
| 5 | 17 (18.1) | 14 (19.4) | 31 (18.7) | 12 (20.7) | 7 (16.7) | 19 (19.0) |
| **Entry route, n (%)** | | | | | | |
| Number with data, n (%) | 105 (96.3) | 77 (93.9) | 182 (95.3) | 64 (100) | 43 (97.8) | 107 (99.1) |
| Caution | 93 (88.6) | 72 (93.5) | 165 (90.7) | 57 (89.1) | 42 (97.7) | 99 (92.5) |
| Prosecution | 12 (11.4) | 5 (6.5) | 17 (9.3) | 7 (10.9) | 1 (2.3) | 8 (7.5) |
| **Total number of RMS incidents involved in 1 year prerandomisation (not including RMS incident that led to study entry)** | | | | | | |
| Number with data, n (%) | 104 (95.4) | 77 (93.9) | 181 (94.8) | 63 (98.4) | 44 (100) | 107 (99.1) |
| Mean (SD) | 10.8 (12.5) | 12.9 (25.7) | 11.7 (19.2) | 9.3 (8.7) | 9.0 (9.9) | 9.2 (9.2) |
| Median (IQR) | 7 (3, 13) | 6 (3, 12) | 6 (3, 13) | 6 (3, 13) | 5 (3, 12) | 6 (3, 13) |
| Min, Max | 0, 79 | 1, 200 | 0, 200 | 0, 35 | 1, 38 | 0, 38 |
| **Total number of RMS incidents leading to charge or caution 1 year prerandomisation (not including charge or caution that led to study entry)** | | | | | | |
| Number with data, n (%) | 104 (95.4) | 77 (93.9) | 181 (94.8) | 63 (98.4) | 44 (100) | 107 (99.1) |
| Mean (SD) | 0.6 (1.0) | 0.5 (1.3) | 0.5 (1.1) | 0.6 (1.0) | 0.3 (0.6) | 0.5 (0.9) |
| Median (IQR) | 0 (0, 1) | 0 (0, 1) | 0 (0, 1) | 0 (0, 1) | 0 (0, 0.5) | 0 (0, 1) |
| Min, Max | 0, 4 | 0, 10 | 0, 10 | 0, 4 | 0, 2 | 0, 4 |
| **Total number of PNC convictions 1 year prerandomisation** | | | | | | |
| Number with data, n (%) | 104 (95.4) | 77 (93.9) | 181 (94.8) | 63 (98.4) | 44 (100) | 107 (99.1) |
| Mean (SD) | 0.5 (0.8) | 0.3 (0.5) | 0.4 (0.7) | 0.4 (0.7) | 0.2 (0.5) | 0.3 (0.6) |
| Median (IQR) | 0 (0, 1) | 0 (0, 1) | 0 (0, 1) | 0 (0, 1) | 0 (0, 0) | 0 (0, 0) |
| Min, Max | 0, 3 | 0, 2 | 0, 3 | 0, 2 | 0, 2 | 0, 2 |
| **Involved in RMS incident that led to caution or charge 1 year prerandomisation (not including charge or caution that led to study entry), n (%)** | | | | | | |
| Number with data, n (%) | 104 (95.4) | 77 (93.9) | 181 (94.8) | 63 (98.4) | 44 (100) | 107 (99.1) |
| Yes | 36 (34.6) | 21 (27.3) | 57 (31.5) | 21 (33.3) | 11 (25.0) | 32 (29.9) |

Continued

**Table 1** Continued

| | Randomised participants (n=191) | | | Provided valid WEMWBS for at least one timepoint (n=108) | | |
|---|---|---|---|---|---|---|
| | Gateway conditional caution (n=109) | Usual process (n=82) | Total (n=191) | Gateway conditional caution (n=64) | Usual process (n=44) | Total (n=108) |
| No | 68 (65.4) | 56 (72.7) | 124 (68.5) | 42 (66.7) | 33 (75.0) | 75 (70.1) |
| PNC conviction 1 year prerandomisation, n (%) | | | | | | |
| Number with data, n (%) | 104 (95.4) | 77 (93.9) | 181 (94.8) | 63 (98.4) | 44 (100) | 107 (99.1) |
| Yes | 31 (29.8) | 22 (28.6) | 53 (29.3) | 16 (25.4) | 8 (18.2) | 24 (22.4) |
| No | 73 (70.2) | 55 (71.4) | 128 (70.7) | 47 (74.6) | 36 (81.8) | 83 (77.6) |

IMD, Index of Multiple Deprivation; PNC, police national computer; RMS, record management system; WEMWBS, Warwick-Edinburgh Mental Well-being Scale.

**Table 2** The Warwick-Edinburgh Mental Well-being Scale score at each timepoint, presented by allocated group

| | Gateway conditional caution (n=109) | Usual process (n=82) |
|---|---|---|
| **Week 4** | | |
| Number with data, n (%) | 57 (52.3) | 36 (43.9) |
| Mean (SD) | 44.1 (9.6) | 44.9 (7.2) |
| Median (IQR) | 45 (38, 52) | 44 (41, 49) |
| Min, Max | 19, 61 | 28, 62 |
| **Week 16** | | |
| Number with data, n (%) | 54 (49.5) | 39 (47.6) |
| Mean (SD) | 48.6 (9.9) | 46.0 (8.5) |
| Median (IQR) | 49 (42, 55) | 47 (40, 53) |
| Min, Max | 27, 67 | 30, 60 |
| **Year 1** | | |
| Number with data, n (%) | 27 (24.8) | 16 (19.5) |
| Mean (SD) | 48.4 (9.7) | 45.7 (7.0) |
| Median (IQR) | 49 (41, 54) | 45.5 (41.5, 50.5) |
| Min, Max | 29, 68 | 28, 58 |

2 or withdraw stage 2 consent, 8 (7.6%) reoffended during the period of the conditional caution. There were 2 (25.0%) participants for whom discretion was applied before taking the decision that they were in breach of the condition not to reoffend. The remaining 6 (75.0%) were referred back to the original investigator. Due to the risk of data disclosure further information is not provided here.

Information on the Index of Multiple Drug Use, adverse childhood experiences and the health economic data are presented in online supplemental appendices C–E, respectively.

## DISCUSSION

The Gateway study is the first RCT in the UK police setting to have a health-related primary outcome requiring consent and individual data collection rather than prioritising criminal justice data on recidivism. We have demonstrated that is possible, using a novel two-stage consent process, to recruit and randomise young people who have committed a minor offence to an RCT in the police setting. OOCDs issued by the police such as conditional cautions for less serious offences have been used in practice for over a decade.[6] Evaluations of such interventions have been carried out, including Cautioning and Relationship Abuse,[9] Checkpoint[5] and Operation Turning Point[9] to assess their impact on recidivism. Our study differed from these examples in that our primary outcome was health related. For ethical reasons therefore we needed participant consent prior to randomisation. A considerable amount of additional work to set up

**Table 3** Secondary and exploratory participant-reported outcomes at each timepoint, presented by allocated group

| | Gateway conditional caution (n=109) | Usual process (n=82) |
|---|---|---|
| **SF-12 mental component** | | |
| Week 4 | | |
| Number with data, n (%) | 57 (52.3) | 36 (43.9) |
| Mean (SD) | 42.4 (12.0) | 43.5 (9.7) |
| Median (IQR) | 43.6 (35.7, 53.1) | 43.8 (36.8, 51.9) |
| Min, Max | 15.1, 58.8 | 22.1, 58.8 |
| Week 16 | | |
| Number with data, n (%) | 54 (49.5) | 39 (47.6) |
| Mean (SD) | 47.7 (7.6) | 45.0 (9.1) |
| Median (IQR) | 47.7 (41.7, 54.6) | 45.8 (38.7, 52.7) |
| Min, Max | 34.3, 58.8 | 20.7, 58.1 |
| Year 1 | | |
| Number with data, n (%) | 27 (24.8) | 16 (19.5) |
| Mean (SD) | 47.5 (7.5) | 46.1 (8.6) |
| Median (IQR) | 47.7 (39.5, 54.6) | 47.5 (44.4, 51.8) |
| Min, Max | 34.3, 58.8 | 20.7, 58.1 |
| **SF-12 physical component** | | |
| Week 4 | | |
| Number with data, n (%) | 57 (52.3) | 36 (43.9) |
| Mean (SD) | 54.5 (5.3) | 52.8 (6.7) |
| Median (IQR) | 55.5 (53.7, 57.4) | 55.2 (51.2, 56.8) |
| Min, Max | 36.8, 63.9 | 30.8, 59.2 |
| Week 16 | | |
| Number with data, n (%) | 54 (49.5) | 39 (47.6) |
| Mean (SD) | 52.5 (6.4) | 53.4 (5.7) |
| Median (IQR) | 54.5 (51.7, 56.0) | 55.2 (52.4, 56.9) |
| Min, Max | 26.1, 59.4 | 38.0, 60.1 |
| Year 1 | | |
| Number with data, n (%) | 27 (24.8) | 16 (19.5) |
| Mean (SD) | 51.9 (7.9) | 53.5 (6.3) |
| Median (IQR) | 54.5 (51.7, 56.5) | 55.3 (52.5, 58.2) |
| Min, Max | 26.1, 59.4 | 38.0, 58.9 |
| **AUDIT** | | |
| Week 4 | | |
| Number with data, n (%) | 57 (52.3) | 36 (43.9) |
| Mean (SD) | 12.9 (9.2) | 11.2 (7.5) |
| Median (IQR) | 11 (5, 19) | 10.5 (5.5, 16.5) |
| Min, Max | 0, 34 | 0, 28 |
| Week 16 | | |
| Number with data, n (%) | 54 (49.5) | 39 (47.6) |
| Mean (SD) | 11.6 (8.1) | 11.6 (8.7) |
| Median (IQR) | 9.5 (5, 15) | 10 (4, 16) |
| Min, Max | 0, 32 | 0, 36 |
| Year 1 | | |

**Table 3** Continued

| | Gateway conditional caution (n=109) | Usual process (n=82) |
|---|---|---|
| Number with data, n (%) | 27 (24.8) | 16 (19.5) |
| Mean (SD) | 11.1 (8.5) | 13.3 (8.3) |
| Median (IQR) | 8 (5, 20) | 12.5 (8, 17) |
| Min, Max | 0, 30 | 1, 30 |
| **ADIS** | | |
| Week 4 | | |
| Number with data, n (%) | 57 (52.3) | 36 (43.9) |
| Mean (SD) | 46.9 (33.6) | 45.1 (36.5) |
| Median (IQR) | 38 (25, 59) | 37.5 (12, 76.5) |
| Min, Max | 0, 137 | 0, 111 |
| Week 16 | | |
| Number with data, n (%) | 54 (49.5) | 39 (47.6) |
| Mean (SD) | 40.9 (36.3) | 37.2 (38.2) |
| Median (IQR) | 36.5 (15, 52) | 31 (0, 67) |
| Min, Max | 0, 137 | 0, 111 |
| Year 1 | | |
| Number with data, n (%) | 27 (24.8) | 16 (19.5) |
| Mean (SD) | 48.7 (36.1) | 50.5 (39.0) |
| Median (IQR) | 40 (23, 68) | 38.5 (20.5, 86) |
| Min, Max | 0, 134 | 0, 111 |
| **Accommodation status (exploratory), n (%)** | | |
| Week 4 | | |
| Number with data, n (%) | 57 (52.3) | 36 (43.9) |
| Homeless | 8 (14.0) | 3 (8.3) |
| Not homeless | 49 (86.0) | 33 (91.7) |
| Year 1, n (%) | | |
| Number with data, n (%) | 27 (24.8) | 15 (18.3) |
| Homeless | 3 (11.1) | 0 (0) |
| Not homeless | 24 (88.9) | 15 (100) |

ADIS, Adolescent Drug Involvement Scale; AUDIT, Alcohol Use Disorders Identification Test; SF-12, Short Form-12.

and for the investigators to administer at a time of stress for potential participants. We were only able to recruit because of the close collaboration between the research team and HC.

A key limitation of the study is that due to high attrition rates, the study was ended early and an assessment of the effectiveness of the Gateway intervention compared with usual process could not be completed. Similar issues with the follow-up and the collection of health data have been found in other community-based studies in disadvantaged populations, especially those with young people.[16 17] We implemented numerous strategies to overcome our issues with retention including a telephone call reminder about the study from the HC Gateway Project Officer before stage 2 consent was due. Our public involvement work with vulnerable young people resulted in valuable

suggestions, which we implemented, including changing the wording on participant facing information and creating a video explaining the study. We also increased the value of the shopping gift cards on offer for return of outcome data. In addition, we put into place strategies to improve recruitment, including expansion of the study catchment area and following up the non-screening of a potentially eligible participant with the recruiting police staff member to ascertain the factors that led to this. However, we were unable to solve the barrier presented by out-of-date or invalid contact details, as well as the lack of response by the participants to contact attempts by the researchers.

The groups were generally well balanced in terms of characteristics and percentage providing data, and allocation did not appear to make any difference to level of

**Table 4** Recidivism outcomes presented by allocated group

| | Gateway conditional caution (n=109) | Usual process (n=82) |
|---|---|---|
| RMS incidents involved in up to 1 year postrandomisation | | |
| Number with data, n (%) | 74 (67.9) | 55 (67.1) |
| Mean (SD) | 9.3 (12.2) | 12.2 (23.7) |
| Median (IQR) | 5 (1, 14) | 5 (1, 11) |
| Min, Max | 0, 61 | 0, 132 |
| Total number of RMS incidents resulting in being classed as a suspect and charged/cautioned up to 1 year postrandomisation | | |
| Number with data, n (%) | 74 (67.9) | 55 (67.1) |
| Mean (SD) | 0.4 (1.2) | 0.8 (2.9) |
| Median (IQR) | 0 (0, 0) | 0 (0, 0) |
| Min, Max | 0, 7 | 0, 20 |
| Total number of PNC convictions up to 1 year postrandomisation | | |
| Number with data, n (%) | 72 (66.1) | 53 (64.6) |
| Mean (SD) | 0.4 (0.8) | 0.4 (0.9) |
| Median (IQR) | 0 (0, 0) | 0 (0, 0) |
| Min, Max | 0, 3 | 0, 5 |
| Charged with a 'summary' or 'either way' offence up to 1 year postrandomisation | | |
| Number with data, n (%) | 72 (66.1) | 53 (63.9) |
| Charged | 19 (26.4) | 16 (30.2) |
| Not charged | 53 (73.6) | 37 (69.8) |
| Charged with an 'indictable only' offence up to 1 year postrandomisation | | |
| Number with data, n (%) | 72 (66.1) | 53 (64.6) |
| Charged | 0 (0) | 0 (0) |
| Not charged | 72 (100) | 53 (100) |

PNC, police national computer; RMS, record management system.

engagement. Participants who took part in data collection interviews completed all parts of the WEMWBS, SF-12, AUDIT and ADIS instruments at all time points. This suggests that the questions were not overly burdensome or intrusive and that telephone interviews were acceptable to those willing to share a valid telephone number.

The challenges in recruiting and retaining participants that we faced, and the strategies we put in place to overcome them will help researchers planning and carrying out future studies with this population. We have also provided a benchmark for attrition in this population and setting, which indicates that further work is needed to identify ways to facilitate engagement between researchers and this vulnerable population.

A regression discontinuity design (RDD) may be a pragmatic solution to the recruitment issues encountered by the Gateway trial,[18] that has been used before in the criminal justice setting.[19 20] The RDD is a quasi-experimental design that allocates participants to intervention or control according to their score on a continuous baseline variable, with the outcome being a continuous measure. If there is no effect of the intervention, then the regression plots of the allocation variable against the outcome of interest will be smooth with no interruption at the point of allocation on the pretest variable. However, if the intervention is effective then there will be a change or discontinuity in the regression slope at the point of allocation.

For example, in the criminal justice setting a prospective RDD could use a standardised offender risk score to assign treatment, with participants scoring above a certain threshold being allocated to the intervention, which is probably more logical and acceptable to staff and offenders than the use of randomisation. A prospective design would allow for outcomes that may not be routinely collected, but are relevant to healthcare professionals and the police, to be collected as part of the study. In theory, the RRD would mitigate against selection bias by assuming that measurement error around the threshold point produces equivalent groups.

## CONCLUSION

We have demonstrated that it is possible to recruit and randomise this study population in a police setting, but recruitment and retention estimates should be conservative. However, more work is needed to identify strategies to improve retention rates when carrying out research with this underserved population.

**Author affiliations**
[1]Department of Health Sciences, University of York, York, UK
[2]University of Southampton, Southampton, UK
[3]Hampshire Constabulary, Southampton, UK
[4]Department of Sociology, University of Southampton Highfield Campus, Southampton, UK
[5]WIHRD, School of Medicine, Southampton University, Southampton, UK
[6]The University of York, York, UK
[7]Public Health Sciences and Medical Statistics, University of Southampton, Southampton, UK

**Acknowledgements** We would like to acknowledge and thank all those who contributed to the study. All the young people who took part in the trial and associated interviews. Gateway Navigators provided by No Limits and Southampton City Council who delivered the intervention; Debbie Willis, Children and Young People's Services Manager and colleagues at The Hampton Trust, responsible for LINX; those organisations taking referrals from the Navigators: The Prince's Trust, Two Saints (Housing) and the Community Mental Health Teams across Hampshire and Isle of Wight, as well as others. Members of the Trial Management Group (TMG) who contributed at various stages of the study: Inspector Benjamin Taylor (Hampshire Constabulary), Rosanna Orlando (Health Economist), Anthony Quinn (Qualitative Researcher), Inspector Stuart Baker (Hampshire Constabulary). Those who supported the study and funded the intervention: Office of Police and Crime Commissioner (PCC) supporting the PCC for Hampshire and Isle of Wight, and senior management at Hampshire Constabulary. All the police investigators who underwent training and recruited participants into the trial. Members of the Public Participation Panel (PPP), and all the members of the public who contributed during the development and delivery of the study. The University of Southampton for being

the study sponsor and to members of the Study Steering and Data Monitoring and Ethics Committee for their advice and support.

**Contributors** AM contributed to the overall study design, wrote the statistical analysis plan, conducted the statistical analysis, contributed to writing and editing the manuscript. AB was a coinvestigator, contributed to conceptualisation and design, funding acquisition, protocol development, and was trial manager for the conduct and delivery of the trial, site setup and data management, manuscript writing and editing. SM was a coinvestigator, contributed to conceptualisation and design, funding acquisition, protocol development, conduct and delivery of the trial and qualitative evaluation, data acquisition, qualitative analysis and manuscript commenting. IW contributed to protocol development and study design, conduct and delivery of the trial and qualitative evaluation, data acquisition, qualitative analysis, manuscript commenting. MB-P was the PPI lead for the study, undertook PPI work, contributed to study design and conduct, quantitative data collection qualitative data collection and analysis, and commenting on the manuscript. CC contributed to protocol development, trial conduct, setting up of sites, data acquisition and checking, commented on the manuscript. AC contributed to protocol development, trial conduct, setting up of sites, data acquisition and processing and commented on the manuscript. EF contributed to the study conduct, project administration, data management, and commented on the manuscript. JF was a coinvestigator contributing to conceptualisation and design, funding acquisition, protocol development, trial conduct, and commenting on the manuscript. CH was a coinvestigator, she contributed to conceptualisation and design, funding acquisition, protocol development, provided oversight of trial conduct and the statistical analysis, and commented on the manuscript. JR as a coinvestigator, contributed to conceptualisation and design, funding acquisition, protocol development, trial conduct, and commented on the manuscript. DT as a coinvestigator, contributed to conceptualisation and design, protocol development, funding acquisition, trial conduct, and commented on the manuscript. LW contributed to project administration, data acquisition, qualitative analysis, manuscript commenting. JP was the chief investigator, and contributed to the conceptualisation and design, funding acquisition, protocol development, trial conduct, manuscript commenting. JP is the study guarantor.

**Funding** The Gateway trial was funded by the National Institute of Health Research (NIHR) Public Health Research Programme (award ID: 16/122/20).

**Competing interests** CH was Deputy Chair of the NIHR HTA commissioning board, NIHR CTU Standing Advisory Committee, HTA Post-Funding Committee teleconference and the HTA Funding Committee Policy Group. JR is a member of the NIHR Editorial Board for HTA and EME. JP is Director of Training, UK Faculty of Public Health.

**Patient and public involvement** Patients and/or the public were involved in the design, or conduct, or reporting, or dissemination plans of this research. Refer to the Methods section for further details.

**Patient consent for publication** Not applicable.

**Ethics approval** This study involves human participants and was approved by University of Southampton Ethics and Research Information Governance Board (ERGO ID: 31911). Participants gave informed consent to participate in the study before taking part.

**Provenance and peer review** Not commissioned; externally peer reviewed.

**Data availability statement** Data are available upon reasonable request.

**ORCID iDs**
Alex Mitchell http://orcid.org/0000-0001-9311-2092
Alison Booth http://orcid.org/0000-0001-7138-6295
James Raftery http://orcid.org/0000-0003-1094-8578

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
