## [Reviewer comments · BMJ Open]

ARTICLE DETAILS

TITLE (PROVISIONAL)	Examining the effectiveness of the Gateway conditional caution on health and wellbeing of young adults committing low-level offences: a randomised controlled trial
AUTHORS	Mitchell, Alex; Booth, Alison; Morgan, Sara; Walker, Inna; Chapman, Caroline; Barlow-Pay, Megan; Cochrane, Ann; Filby, Emma; Fleming, Jenny; Hewitt, Catherine; Raftery, James; Torgerson, David; Weir, Lana; Parkes, Julie

VERSION 1 – REVIEW

REVIEWER	Stoliker, Bryce E. University of Saskatchewan, Centre for Forensic Behavioural Science and Justice Studies
REVIEW RETURNED	21-Nov-2023

GENERAL COMMENTS	This study provides a descriptive and exploratory analysis of health, social, and criminal justice outcomes across various time points (4, 16, 52 weeks) for young adults who committed a low-level offence and were randomly assigned to a rehabilitative program (i.e., the Gateway conditional caution) vs. usual criminal justice process. While the study and its findings are interesting, the manuscript requires major revision. Specifically, greater clarity is needed in terms of the description of the study methodology, and greater attention should be placed on the key outcomes and findings (along with relevant implications for research and practice) as opposed to study protocol. I have detailed specific comments, questions, and recommendations below according to each section of the main manuscript. General Comments: - The authors indicate throughout, leading up to the end of the Methods section, that the study examined the effectiveness of the Gateway program, but suggest that due to attrition and data collection issues, this could not be examined, and only descriptive analyses are provided according to groups (i.e., treatment vs. control). The narrative around the aim and purpose of the study should be revised to better reflect its strictly descriptive and exploratory nature, as opposed to presenting it as a randomized control trial that examined effectiveness of treatment vs. control (as this did not occur).- Overall, the narrative for the manuscript emulates more of a study protocol than discussion on data and findings related to the study in question. At the beginning of the manuscript, focus is on how the study assesses the effectiveness of the Gateway program compared with usual processes, but in the Methods, Results, and Discussion sections the focus is primarily on study protocol
---

including aspects of retention/recruitment, with limited focus on the key outcomes and how the program (vs. usual process) impacts the various key outcomes.

BACKGROUND

- Pg. 6, line 8-15: The authors state that “these young adults” are more likely to come into contact with police as suspects/victims and are overrepresented in the CJS. Are the authors referring to young adults with low-level offences specifically, or young adults with high level health, social, and mental health needs?

- Pg. 6, line 29-34: The authors end this sentence with “...instead of entering the criminal justice system.” Wouldn’t the individuals still be involved in the criminal justice system, just in the capacity of a rehabilitative community program? So, perhaps referring to the fact that they do not enter less rehabilitative pathways, such as prison?

- Pg. 6, line 34-36: What is meant by the term “out-of-court disposals”? It would be good to explain, even in just a footnote, for international audiences.

- Pg. 6, line 38: Arguably, the aim would more likely to be to divert the young adult away from the formal criminal justice system to better address their offending behaviour through more rehabilitative pathways.

- Pg. 6, line 41: As above, what does “conditional caution” refer to for an international audience?

- Pg. 6, line 41-50: The authors introduce the program under investigation for this study, which appears to be a program that existed outside of this research. In that case, when was the program established and/or how long had it been operating? Relatedly, are there any details on the number of individuals who have received services through the program as it operated outside of the current study?

METHODS

- Pg. 7, line 28-30: When the authors refer to the recruitment of participants from four sites, what are the sites in question? That is, police detachments or headquarters, courts, etc.?

- Pg. 7, line 31-33: The authors highlight the timeline for follow-up; however, what took place in terms of baseline assessment. That is, how were participants assessed at baseline as they “entered” the study?

- Pg. 7, line 38-50: The authors note that exclusion criteria include “breach of offence orders.” This made me wonder another potential aspect of inclusionary/exclusionary criteria the authors could note. Specifically, whether participants include those who have come into contact with the formal criminal justice system for the first time (i.e., first-time offence/index offence), have had a history of offences, or both? Relatedly, it’d be interesting to know, if participants include those with a history of offences, do they only have a history of low-level offences or do that have a history of more serious offences and this particular contact with the criminal justice system is for a low-level offence?

	- Pg. 8, line 45-49: The authors state the following, however, it's unclear what this means exactly in terms of study procedure: "It was therefore possible that the subsequent in person disposal for some of these participants could occur several weeks after randomisation depending on when the in-person disposal could be arranged." - Pg. 8, line 52-59 and pg. 9, line 3-8: The authors describe the consent process and data collection time points, referring to follow-up time periods. Were baseline data collected on the sample at any point? That is, prior to entering the treatment or control groupings and exposures. - Pg. 9, line 13-20: The authors refer to training for police officers/investigators. What was the purpose of the training? Was it to train them on study procedures (i.e., screening for eligible participants, recruitment, etc.)? - Pg. 9, line 20-22: The authors note for the first time that an eligibility tool was used to screen for potential participants. What was the basis of this tool and was it used by police investigators? Prior to this note, it was implied that police screened for eligibility based on (what could reasonably be assumed) to be criteria generally provided by the researchers, but with no standardized tool. Further clarification on the screening processes for eligible participants is therefore needed prior to, and alongside, this sentence. - Pg. 9, line 27-29: How was it ensured that researchers involved in consenting and data collection were blind to participants' allocation to intervention vs. control group? - General comments related to recruitment and randomization: (1) Do the authors have any information on the number of total eligible participants that could have been involved in Gateway and/or the study prior to the final sample (i.e., what number of individuals were recruited, or could have been recruited, by police investigators) to provide an idea of the sampling frame vs. final sample. (2) How did study participants "enter" the intervention vs. control groups. That is, did the intervention-based participants all start the intervention at the same time, or on a rolling basis? Same for control group (i.e., treatment as usual). (3) Were participants first consented by investigators, then randomized, then consented by researchers to be a part of the study? (4) It is not entirely clear what the process is for allocation to the intervention vs. control groups (i.e., randomization). The authors should provide a clearer description of how exactly participants were allocated to the treatment vs. control groups for the study. For instance, what is a 1:1 allocation ratio? Does this mean alternating participants to groups as they were recruited? Relatedly, how were participants recruited for the intervention vs. control groups? Were certain participants "more eligible" for the Gateway vs. usual process? In this case, did police investigators present the opportunity for Gateway and/or the usual process (as well as the possibility of participating in the study) to all prospective participants, or were some prospective participants given the option of Gateway whereas others were not? - Pg. 9, line 54: It's noted that Gateway navigators were trained practitioners. What kind of practitioners? For instance, social
--	---

	workers, mental health professionals, criminal justice professionals, etc.?  - Pg. 9-10: Under the subheading “Intervention and usual care” the authors describe the compulsory elements of the Gateway intervention. However, there doesn’t seem to be a description of what the intervention provides on a broad level, as well as on a day-to-day basis, for individuals who receive these services. The authors should provide a description and/or cite to another paper that provides greater detail on the intervention. - The author(s) don’t mention the study period (i.e., study start and end date), which is relevant given the mention of the impact of COVID-19 on pg. 10. - The mention of impact of COVID-19 is particularly relevant as criminal justice systems in many countries addressed charges, especially low-level offences, differently (i.e., lower likelihood of incarceration to reduce the spread of COVID-19 where possible). Not to say it is the same for the region of relevance to the current study, but it could have an impact on how cases were managed in addition to the obvious restrictions to in-person activities which impacted things like programming access (e.g., Gateway) and court proceedings. Thus, COVID-19 could have impacted general recruitment of participants on the basis of eligibility, impacting study participation and allocation. - Pg 11, like 28-33: There’s not much detail in terms of the characteristics of the scales used and how scores are interpreted. - Pg. 11, line 37-58 and pg. 12, line 3-6: Related to the recidivism data, the authors should clarify the following: (1) what types of RMS incidents were included? Did this include breaches of conditions (i.e., administrative offences) or a completely new index offence, or a combination? Does it also include general police interactions unrelated to an offence or contacts due to follow up on the individual in some capacity (and thus not really aligning with the definition of recidivism)?; (2) do each of the elements described by the authors represent a different category of data that were examined on the basis of recidivism (e.g., total number of RMS incidents is one category, total number of RMS incidents leading to charge or caution, total number of PNC convictions, etc.)? - More detail could be provided on the study outcome measures. For instance, validity and reliability based on previous evidence (and relevant citations). Any use with similar samples in previous studies (and citations). Examples of items asked. Etc. - Overall, there’s not much information provided in the Methods section in terms of participant numbers from point of recruitment, to allocation to intervention vs. control, to attrition/retention. Relatedly, what was the level of attrition for the treatment vs. control groups. Generally, more information is needed on the sampling frame and procedure, and how the authors came to the final sample (and further how retention/attrition impacted the sample over the study period). - Pg. 13, line 21-28: The authors discuss statistical techniques and approaches for the WEMWBS, but don’t provide any context to
--	--

	this discussion or what exactly it means in terms of the current study and analyses. - Pg. 13, line 28-32: The authors note the number of participants required for statistical power. Is this the total number of participants for the study sample to detect differences between the two groups (i.e., treatment vs. control) or to detect changes in scores from one timepoint to another (i.e., repeated measures). - Pg. 13, line 32-37: What was the justification for using the RADAR intervention as a model to inform attrition for the current study? Is it a similar intervention or, at the very least, a similar type of model that the Gateway uses? Also, is this RADAR attrition rate referring to program-only attrition or study-based attrition (i.e., people who drop out of the program itself, or attrition based on both treatment and control groups according to a study on this intervention)? - How long is the Gateway intervention? For instance, are participants involved in the program at 4, 16, 1 year? Or were those time points that assessed participants following completion of the intervention? This would be relevant to interpreting the findings in terms of whether outcomes, such as mental well-being, increased over time especially if they were involved in the program from point 0 to 1 year later; thus, longer involvement, increases over time would be expected. Alternatively, it would also help explain a sharp increase in weeks following the intervention, followed by a tapering effect and the effects of interventions may not last over time. Either way, knowing the duration of the programming and whether participants were assessed while involved in the programming is important to better understand the findings. RESULTS - Pg. 15, line 11-14: It is noted when recruitment and data collection ended, but there's no clear indication of recruitment/data collection start date for the study (which should be highlighted in the Method section). - Pg. 15, line 41: When referring to the median number of RMS incidents, is this the median for the total sample or number per individual? - Pg. 18, line 22-27: The number of successful contacts is mentioned, but by what standards is an acceptable number of contacts. And what is too low? There's no comparison as to what is exceptional, good, poor, bad, etc. Same thing for duration of successful contacts. - Overall, the Results sections provides a heavy amount of discussion on various participant groups and subgroups across the time points of the study, but it's (a) difficult to follow and (b) unclear where all these participant numbers are coming from. It also detracts from the main findings. Very little time is spent on highlighting the key primary, secondary, and tertiary outcomes. Most time is spent on describing the number of participants. While describing the groupings and subgroupings across time points is relevant, it again detracts from the findings when focusing primarily on this. It would be most effective to reflect on what the findings
--	---

	show for key outcomes. For instance, do scores go up, down, stay the same? - It's not clear why the authors did not conduct between and/or within group-based comparisons on outcome data, even just for one timepoint (but ideally all). While the number of participants isn't high, it's not incredibly low either, and statistical tests are still feasible. Especially when looking at key outcomes, like WEMWBS, SF-12, AUDIT, ADIS, and (some) recidivism data. For instance, using Multivariate ANOVA, repeated measures ANOVA, one-way ANOVA, independent samples t-test, paired samples t-test, etc. (whatever may be most appropriate for the data/test). In any case, group-based comparisons, even if exploratory at this stage, would provide a better understanding of the true differences between treatment and control on the various (key) outcomes. DISCUSSION - The Discussion section will likely require revision to reflect the abovementioned revisions. - Pg. 22, line 33-44: As above, the authors note the focus of the study being a randomized control trial. However, the only component that followed for this was the group assignment (though, information on group assignment is still limited). The analyses did not follow the assessment of treatment vs. control to determine group-based differences on key outcomes. - The Discussion section is quite limited, and is primarily lacking in terms of highlighting (a) comparisons between the current study findings and previous work, and (b) major implications based on the key findings. Currently, the Discussion section is research-oriented, which is relevant, but there seems to be no mention of practical implications of the findings.
--	--

REVIEWER	Neyroud, Peter University of Cambridge
REVIEW RETURNED	05-Dec-2023

GENERAL COMMENTS	This article described the evaluation of a potentially important trial of a conditional caution intervention for young adults. The article makes the claim that it is reporting the first UK based RCT that has tested an intervention with a health-based outcome measure rather than a reliance on standard criminal justice measures of reoffending (prevalence, frequency and harm). That is quite a significant claim and one of many reasons why there was a good case for reporting a trial that did not succeed because it failed to recruit a sufficient number of subjects to meet the protocol target. However, to strengthen that claim the reference list and literature reviewed seems light. The Campbell systematic reviews cited are both focused on younger offenders than the young adult group studied here and there is no attempt to survey the development of conditional caution and deferred prosecutions in the UK. Had they done so the authors would have found quite a lot of material (Turning Point, CARA and Checkpoint for example) which would also have provided them with points of reference for some of the operational difficulties that they confronted as the research progressed. Operation Turning Point (London) may not have fully reported yet, but it managed to run through COVID without some of the issues that this research encountered.
--

	Indeed, because the authors have not felt able to report fully on the results because of the attrition from their sample, the article has, effectively, become more akin to an implementation study reporting the issues that they encountered in a novel study and the methods that used to try to overcome them. As such, it is still very much worthy of publication. The article ends up documenting a fascinating intersection between health and criminal justice research approaches. One issue that could have been more clearly set out is the comparison between the control treatments and the Gateway treatment. The text hinted that the control offenders may have had a range of treatments and conditions but a Table with these set out and compared to the Gateway would have made this clearer. There was one really grating issue in reading the article and that is the overuse of acronyms. Whilst policing is a landscape rich in acronyms, the article makes heavy and unnecessary use of them and then provides an inadequate glossary which does not include many of the terms used. The article would read better with fewer acronyms and a better glossary for the ones used.
--	--

VERSION 1 – AUTHOR RESPONSE

Reviewer 1

- The authors indicate throughout, leading up to the end of the Methods section, that the study examined the effectiveness of the Gateway program, but suggest that due to attrition and data collection issues, this could not be examined, and only descriptive analyses are provided according to groups (i.e., treatment vs. control). The narrative around the aim and purpose of the study should be revised to better reflect its strictly descriptive and exploratory nature, as opposed to presenting it as a randomized control trial that examined effectiveness of treatment vs. control (as this did not occur).

We believe that in terms of transparency it is important to make it clear that this study was designed as a full-scale randomised controlled trial assessing effectiveness. This was the aim and purpose of the study set out in the publicly available protocol. We do make it clear in the abstract, methods and discussion section that it was not feasible to assess the effectiveness of the intervention.

- Overall, the narrative for the manuscript emulates more of a study protocol than discussion on data and findings related to the study in question. At the beginning of the manuscript, focus is on how the study assesses the effectiveness of the Gateway program compared with usual processes, but in the Methods, Results, and Discussion sections the focus is primarily on study protocol including aspects of retention/recruitment, with limited focus on the key outcomes and how the program (vs. usual process) impacts the various key outcomes.

We have reported our trial in accordance with the CONSORT reporting guidelines. At the beginning of the manuscript, we state the aim of the study. Unfortunately, due to low retention rates it was not feasible or appropriate to assess the effectiveness of the intervention in terms of the planned outcomes. Therefore we focussed on reporting recruitment and retention data and discussing measures used to try and overcome issues encountered. It is important to share the results of trials,

whether successful or not and whether findings are positive or negative. We anticipate that sharing our learning from running this trial will benefit future research in this area.

- Pg. 6, line 8-15: The authors state that “these young adults” are more likely to come into contact with police as suspects/victims and are overrepresented in the CJS. Are the authors referring to young adults with low-level offences specifically, or young adults with high level health, social, and mental health needs?

We have changed “These young adults...” to “These young offenders...”. In addition we have referenced <https://revolving-doors.org.uk/wp-content/uploads/2022/02/1309-Broke-but-not-broken-Reportv2-3.pdf>.

- Pg. 6, line 29-34: The authors end this sentence with “...instead of entering the criminal justice system.” Wouldn’t the individuals still be involved in the criminal justice system, just in the capacity of a rehabilitative community program? So, perhaps referring to the fact that they do not enter less rehabilitative pathways, such as prison?

We have amended the wording to “a court summons”.

- Pg. 6, line 34-36: What is meant by the term “out-of-court disposals”? It would be good to explain, even in just a footnote, for international audiences.

We have added the wording “(an alternative to a court summons)”.

- Pg. 6, line 38: Arguably, the aim would more likely be to divert the young adult away from the formal criminal justice system to better address their offending behaviour through more rehabilitative pathways.

We have added the wording “through a rehabilitative path.”

- Pg. 6, line 41: As above, what does “conditional caution” refer to for an international audience?

We have added the wording “where release from custody comes with agreed conditions.”

- Pg. 6, line 41-50: The authors introduce the program under investigation for this study, which appears to be a program that existed outside of this research. In that case, when was the program established and/or how long had it been operating? Relatedly, are there any details on the number of individuals who have received services through the program as it operated outside of the current study?

The Gateway programme brought together elements that had previously been used in isolation. The programme was being set up at the same time as the study. We have amended the wording from “lack of” to “need for” evidence.

- Pg. 7, line 28-30: When the authors refer to the recruitment of participants from four sites, what are the sites in question? That is, police detachments or headquarters, courts, etc.?

We have added the words "Police Stations".

- Pg. 7, line 31-33: The authors highlight the timeline for follow-up; however, what took place in terms of baseline assessment. That is, how were participants assessed at baseline as they "entered" the study?

It was not possible to collect baseline data for the primary and secondary outcomes prior to randomisation because of the required consenting process and statutory police requirements. Randomisation had to take place prior to disposal with the intervention starting within a few days of this. We therefore elected instead to collect outcome data at three time points, Week 4 during the intervention; week 16 at the end of the intervention delivery and at 1 year post randomisation.

- Pg. 7, line 38-50: The authors note that exclusion criteria include "breach of offence orders." This made me wonder another potential aspect of inclusionary/exclusionary criteria the authors could note. Specifically, whether participants include those who have come into contact with the formal criminal justice system for the first time (i.e., first-time offence/index offence), have had a history of offences, or both? Relatedly, it'd be interesting to know, if participants include those with a history of offences, do they only have a history of low-level offences or do that have a history of more serious offences and this particular contact with the criminal justice system is for a low-level offence?

In Table 1 we summarise the number of RMS incidents and PNC convictions one-year pre-randomisation, with PNC convictions being a proxy measure for more serious offences.

- Pg. 8, line 45-49: The authors state the following, however, it's unclear what this means exactly in terms of study procedure: "It was therefore possible that the subsequent in person disposal for some of these participants could occur several weeks after randomisation depending on when the in-person disposal could be arranged."

We have added the words "Study procedures continued as per protocol."

- Pg. 8, line 52-59 and pg. 9, line 3-8: The authors describe the consent process and data collection time points, referring to follow-up time periods. Were baseline data collected on the sample at any point? That is, prior to entering the treatment or control groupings and exposures.

It was not possible to collect baseline data for the primary and secondary outcomes prior to randomisation because of the required consenting process. Randomisation had to take place prior to disposal with the intervention starting within a few days of this.

- Pg. 9, line 13-20: The authors refer to training for police officers/investigators. What was the purpose of the training? Was it to train them on study procedures (i.e., screening for eligible participants, recruitment, etc.)?

In the section on Recruitment we state that “HC investigators were trained to identify, recruit and randomise participants, an approach that had previously been used.” We have in addition amended the wording referred to above to show that the investigators were “provided with training in study procedures”.

- Pg. 9, line 20-22: The authors note for the first time that an eligibility tool was used to screen for potential participants. What was the basis of this tool and was it used by police investigators? Prior to this note, it was implied that police screened for eligibility based on (what could reasonably be assumed) to be criteria generally provided by the researchers, but with no standardized tool. Further clarification on the screening processes for eligible participants is therefore needed prior to, and alongside, this sentence.

We have added the words ‘by the investigators’. Prior references in the manuscript to investigator screening relate to identifying “potentially eligible participants” in order to obtain stage 1 consent. The eligibility tool was not used until consent had been given.

- Pg. 9, line 27-29: How was it ensured that researchers involved in consenting and data collection were blind to participants’ allocation to intervention vs. control group?

Information about allocation was not shared with the University of Southampton where the research involved in consenting and data collection were based. Details are provided in the Protocol paper which is referenced and in the protocol which is publicly available.

- General comments related to recruitment and randomization: (1) Do the authors have any information on the number of total eligible participants that could have been involved in Gateway and/or the study prior to the final sample (i.e., what number of individuals were recruited, or could have been recruited, by police investigators) to provide an idea of the sampling frame vs. final sample.

This information is provided in the results section.

(2) How did study participants “enter” the intervention vs. control groups. That is, did the intervention-based participants all start the intervention at the same time, or on a rolling basis? Same for control group (i.e., treatment as usual).

As we state in the description of the intervention, participants met with a Gateway navigator within 3-5 working days of their disposal. Likewise usual process followed individual timelines, as per standard practice.

(3) Were participants first consented by investigators, then randomized, then consented by researchers to be a part of the study?

Yes. Please see the section on Recruitment

(4) It is not entirely clear what the process is for allocation to the intervention vs. control groups (i.e., randomization). The authors should provide a clearer description of how exactly participants were allocated to the treatment vs. control groups for the study. For instance, what is a 1:1 allocation ratio? Does this mean alternating participants to groups as they were recruited? Relatedly, how were participants recruited for the intervention vs. control groups? Were certain participants “more eligible” for the Gateway vs. usual process? In this case, did police investigators present the opportunity for Gateway and/or the usual process (as well as the possibility of participating in the study) to all prospective participants, or were some prospective participants given the option of Gateway whereas others were not?

We have reworded the section on Randomisation and blinding to address the points about allocation.

We have clarified that when Investigators identified potentially eligible participants they discussed their options, including the Gateway caution. All young person meeting the eligibility criteria were equally eligible to take part in the study and each had an equal chance of being randomised to either group.

- Pg. 9, line 54: It's noted that Gateway navigators were trained practitioners. What kind of practitioners? For instance, social workers, mental health professionals, criminal justice professionals, etc.?

We have replaced “practitioners” with “support workers”.

- Pg. 9-10: Under the subheading “Intervention and usual care” the authors describe the compulsory elements of the Gateway intervention. However, there doesn't seem to be a description of what the intervention provides on a broad level, as well as on a day-to-day basis, for individuals who receive these services. The authors should provide a description and/or cite to another paper that provides greater detail on the intervention.

Full details of the intervention and usual care are provided in the protocol and protocol paper.

- The author(s) don't mention the study period (i.e., study start and end date), which is relevant given the mention of the impact of COVID-19 on pg. 10.

This information is stated at the start of the results section.

- The mention of impact of COVID-19 is particularly relevant as criminal justice systems in many countries addressed charges, especially low-level offences, differently (i.e., lower likelihood of incarceration to reduce the spread of COVID-19 where possible). Not to say it is the same for the region of relevance to the current study, but it could have an impact on how cases were managed in addition to the obvious restrictions to in-person activities which impacted things like programming access (e.g., Gateway) and court proceedings. Thus,

COVID-19 could have impacted general recruitment of participants on the basis of eligibility, impacting study participation and allocation.

Thank you for raising this point.

- Pg 11, like 28-33: There's not much detail in terms of the characteristics of the scales used and how scores are interpreted.

We feel this level of detail is unnecessary given no analyses were carried out. Further details are available in the study protocol paper which we reference.

- Pg. 11, line 37-58 and pg. 12, line 3-6: Related to the recidivism data, the authors should clarify the following: (1) what types of RMS incidents were included? Did this include breaches of conditions (i.e., administrative offences) or a completely new index offence, or a combination? Does it also include general police interactions unrelated to an offence or contacts due to follow up on the individual in some capacity (and thus not really aligning with the definition of recidivism)?; (2) do each of the elements described by the authors represent a different category of data that were examined on the basis of recidivism (e.g., total number of RMS incidents is one category, total number of RMS incidents leading to charge or caution, total number of PNC convictions, etc.)?

(1) As stated RMS incidents included were those resulting in being charged or cautioned. Therefore if a breach of a conditional caution occurred and it resulted in charges being brought it would have been included. We believe that 'resulting in being charged or cautioned' is a clear enough statement to rule out any other forms of interaction.

(2) As we explain at the top of page 12, on receipt of the data there was a clear risk of double counting, therefore we present the RMS and PNC data separately.

- More detail could be provided on the study outcome measures. For instance, validity and reliability based on previous evidence (and relevant citations). Any use with similar samples in previous studies (and citations). Examples of items asked. Etc.

We feel adding this level of detail is unnecessary given no analyses were carried out. Further details are available in the study protocol paper which we reference.

- Overall, there's not much information provided in the Methods section in terms of participant numbers from point of recruitment, to allocation to intervention vs. control, to attrition/retention. Relatedly, what was the level of attrition for the treatment vs. control groups. Generally, more information is needed on the sampling frame and procedure, and how the authors came to the final sample (and further how retention/attrition impacted the sample over the study period).

Details on the recruitment process are provided in the methods section under 'Recruitment', and specific figures are provided at the start of the results section.

- Pg. 13, line 21-28: The authors discuss statistical techniques and approaches for the WEMWBS, but don't provide any context to this discussion or what exactly it means in terms of the current study and analyses.

In line with required reporting for the Consort statement. The planned analyses were not carried out due to lack of data/missing data, negating the need for further discussion.

- Pg. 13, line 28-32: The authors note the number of participants required for statistical power. Is this the total number of participants for the study sample to detect differences between the two groups (i.e., treatment vs. control) or to detect changes in scores from one timepoint to another (i.e., repeated measures).

As we were unable to obtain baseline data, we collected outcome data at three time points with the intentions of undertaking a repeated measures analysis.

- Pg. 13, line 32-37: What was the justification for using the RADAR intervention as a model to inform attrition for the current study? Is it a similar intervention or, at the very least, a similar type of model that the Gateway uses? Also, is this RADAR attrition rate referring to program-only attrition or study-based attrition (i.e., people who drop out of the program itself, or attrition based on both treatment and control groups according to a study on this intervention)?

The RADAR intervention was the closest match to Gateway at the time, and was delivered by The Hampton Trust, who also delivered the LINX workshops for Gateway. As we state, the rates were from the intervention; hence the conservative approach to attrition.

- How long is the Gateway intervention? For instance, are participants involved in the program at 4, 16, 1 year? Or were those time points that assessed participants following completion of the intervention? This would be relevant to interpreting the findings in terms of whether outcomes, such as mental well-being, increased over time especially if they were involved in the program from point 0 to 1 year later; thus, longer involvement, increases over time would be expected. Alternatively, it would also help explain a sharp increase in weeks following the intervention, followed by a tapering effect and the effects of interventions may not last over time. Either way, knowing the duration of the programming and whether participants were assessed while involved in the programming is important to better understand the findings.

As stated in the methods section, the Gateway intervention lasted for the 16 weeks of the conditional caution (Page 11). Data collection was timed for during (4 weeks) at the end of the intervention (16 weeks) and at one year post randomisation.

- Pg. 15, line 11-14: It is noted when recruitment and data collection ended, but there's no clear indication of recruitment/data collection start date for the study (which should be highlighted in the Method section).

This is stated at the start of the results section 'Between the 1st of October 2019 and 13th December 2021...'

- Pg. 15, line 41: When referring to the median number of RMS incidents, is this the median for the total sample or number per individual?

The number of RMS incidents was calculated for each participant, and the median was taken of this set of numbers.

- Pg. 18, line 22-27: The number of successful contacts is mentioned, but by what standards is an acceptable number of contacts. And what is too low? There's no comparison as to what is exceptional, good, poor, bad, etc. Same thing for duration of successful contacts.

We agree with the reviewer's point, however given this is still a relatively under-researched study population, there are limited benchmarks for comparison.

- Overall, the Results sections provides a heavy amount of discussion on various participant groups and subgroups across the time points of the study, but it's (a) difficult to follow and (b) unclear where all these participant numbers are coming from. It also detracts from the main findings. Very little time is spent on highlighting the key primary, secondary, and tertiary outcomes. Most time is spent on describing the number of participants. While describing the groupings and subgroupings across time points is relevant, it again detracts from the findings when focusing primarily on this. It would be most effective to reflect on what the findings show for key outcomes. For instance, do scores go up, down, stay the same?

We reported the number of participants at each timepoint in order to demonstrate that a high proportion of participants were lost to follow-up. This means that any attempt at assessing the effectiveness of the intervention would be at high risk of bias due to missing data. We summarise the outcomes descriptively in order to meet CONSORT reporting guidelines.

- It's not clear why the authors did not conduct between and/or within group-based comparisons on outcome data, even just for one timepoint (but ideally all). While the number of participants isn't high, it's not incredibly low either, and statistical tests are still feasible. Especially when looking at key outcomes, like WEMWBS, SF-12, AUDIT, ADIS, and (some) recidivism data. For instance, using Multivariate ANOVA, repeated measures ANOVA, one-way ANOVA, independent samples t-test, paired samples t-test, etc. (whatever may be most appropriate for the data/test). In any case, group-based comparisons, even if exploratory at this stage, would provide a better understanding of the true differences between treatment and control on the various (key) outcomes.

We state in the statistical analysis section that due to poor retention and data collection rates, it was not feasible to assess the effectiveness of the Gateway intervention. Therefore, it would be inappropriate to carry out formal hypothesis testing.

- The Discussion section will likely require revision to reflect the abovementioned revisions.

- Pg. 22, line 33-44: As above, the authors note the focus of the study being a randomized control trial. However, the only component that followed for this was the group assignment

(though, information on group assignment is still limited). The analyses did not follow the assessment of treatment vs. control to determine group-based differences on key outcomes.

The study design was that of a randomised controlled trial. The analysis of a randomised controlled trial is separate from the design and can take a range of forms. For example, it would in general be inappropriate to analyse a pilot randomised controlled trial using formal hypothesis testing, as such a study would likely be underpowered to detect a difference. In this case the analysis may instead focus on recruitment and retention rates in order to assess if the study is feasible.

- The Discussion section is quite limited, and is primarily lacking in terms of highlighting (a) comparisons between the current study findings and previous work, and (b) major implications based on the key findings. Currently, the Discussion section is research-oriented, which is relevant, but there seems to be no mention of practical implications of the findings.

Given that we were unable to assess the effectiveness of the Gateway intervention, it would not be inappropriate to discuss practical implications. There are however many research related lessons.

Reviewer 2

However, to strengthen that claim the reference list and literature reviewed seems light. The Campbell systematic reviews cited are both focused on younger offenders than the young adult group studied here and there is no attempt to survey the development of conditional caution and deferred prosecutions in the UK. Had they done so the authors would have found quite a lot of material (Turning Point, CARA and Checkpoint for example) which would also have provided them with points of reference for some of the operational difficulties that they confronted as the research progressed. Operation Turning Point (London) may not have fully reported yet, but it managed to run through COVID without some of the issues that this research encountered.

We thank the reviewer for highlighting these points. We were aware of Turning Point, CARA and Checkpoint. Inspector Andrew Crowe, an author on the Checkpoint study, was a member of our study advisory panel. The use of CARA in Hampshire meant some of our otherwise potentially eligible participants were excluded from being approached. The fact that CARA had successfully trained and used police officers to recruit to that trial supported our approach (Reference 12). Our study police colleagues kept us informed of progress with Operation Turning Point, attending presentations by that team, and sharing information about the Gateway study. When the issuing of conditional cautions was halted by Hampshire Constabulary during COVID, we had no option but to suspend recruitment to the trial.

Our study differed from these examples in that our primary outcome was health related, rather than recidivism, and we aimed to collect both primary and secondary outcome data from both groups. For ethical reasons therefore we needed participant consent prior to randomisation, a not inconsiderable amount of work which neither Checkpoint or Operation Turning Point needed.

One issue that could have been more clearly set out is the comparison between the control treatments and the Gateway treatment. The text hinted that the control offenders may have had

a range of treatments and conditions but a Table with these set out and compared to the Gateway would have made this clearer.

Thank you, Table 1 in Appendix A of the supplementary materials outlines the conditions attached to cautions in both the intervention and control groups.

There was one really grating issue in reading the article and that is the overuse of acronyms. Whilst policing is a landscape rich in acronyms, the article makes heavy and unnecessary use of them and then provides an inadequate glossary which does not include many of the terms used. The article would read better with fewer acronyms and a better glossary for the ones used.

Thank you for pointing out the inaccuracies in the glossary, we have now updated this. While we accept the point about the use of acronyms, the use of them was necessary to fall under the word count required of the submission.

VERSION 2 – REVIEW

REVIEWER	Stoliker, Bryce E. University of Saskatchewan, Centre for Forensic Behavioural Science and Justice Studies
REVIEW RETURNED	28-Mar-2024
GENERAL COMMENTS	The authors have adequately addressed comments/concerns from the original review.